# Systematic Qualitative and Quantitative Analyses of Wenxin Granule via Ultra-High Performance Liquid Chromatography Coupled with Ion Mobility Quadrupole Time-of-Flight Mass Spectrometry and Triple Quadrupole–Linear Ion Trap Mass Spectrometry

**DOI:** 10.3390/molecules27113647

**Published:** 2022-06-06

**Authors:** Yueguang Mi, Wandi Hu, Weiwei Li, Shiyu Wan, Xiaoyan Xu, Meiyu Liu, Hongda Wang, Quanxi Mei, Qinhua Chen, Yang Yang, Boxue Chen, Meiting Jiang, Xue Li, Wenzhi Yang, Dean Guo

**Affiliations:** 1State Key Laboratory of Component-Based Chinese Medicine, Tianjin Key Laboratory of TCM Chemistry and Analysis, Tianjin University of Traditional Chinese Medicine, 10 Poyanghu Road, Jinghai, Tianjin 301617, China; miyueguang@163.com (Y.M.); hwdcrown@163.com (W.H.); lww11413@163.com (W.L.); xxy_0421@163.com (X.X.); lmydz1999@163.com (M.L.); 17862987156@163.com (H.W.); cbx1026tju@163.com (B.C.); jiangmeiting21@163.com (M.J.); tjdxsyx@163.com (X.L.); gda5958@163.com (D.G.); 2Shenzhen Baoan Authentic TCM Therapy Hospital, Shenzhen 518101, China; wanshiyuaaa@163.com (S.W.); meiquanxi@163.com (Q.M.); cqh77@163.com (Q.C.); yangyanghb@outlook.com (Y.Y.); 3Shanghai Research Center for Modernization of Traditional Chinese Medicine, National Engineering Laboratory for TCM Standardization Technology, Shanghai Institute of Materia Medica, Chinese Academy of Sciences, 501 Haike Road, Shanghai 201203, China

**Keywords:** Wenxin granule, multicomponent characterization, HDMS^E^-HDDDA, UHPLC-sMRM, quality evaluation

## Abstract

Wenxin granule (WXG) is a popular traditional Chinese medicine (TCM) preparation for the treatment of arrhythmia disease. Potent analytical technologies are needed to elucidate its chemical composition and assess the quality differences among multibatch samples. In this work, both a multicomponent characterization and quantitative assay of WXG were conducted using two liquid chromatography–mass spectrometry (LC-MS) approaches. An ultra-high performance liquid chromatography–ion mobility quadrupole time-of-flight mass spectrometry (UHPLC/IM-QTOF-MS) approach combined with intelligent peak annotation workflows was developed to characterize the multicomponents of WXG. A hybrid scan approach enabling alternative data-independent and data-dependent acquisitions was established. We characterized 205 components, including 92 ginsenosides, 53 steroidal saponins, 14 alkaloids, and 46 others. Moreover, an optimized scheduled multiple reaction monitoring (sMRM) method was elaborated, targeting 24 compounds of WXG via ultra-high performance liquid chromatography–triple quadrupole linear ion trap mass spectrometry (UHPLC/QTrap-MS), which was validated based on its selectivity, precision, stability, repeatability, linearity, sensitivity, recovery, and matrix effect. By applying this method to 27 batches of WXG samples, the content variations of multiple markers from Notoginseng Radix et Rhizoma (21) and Codonopsis Radix (3) were depicted. Conclusively, we achieved the comprehensive multicomponent characterization and holistic quality assessment of WXG by targeting the non-volatile components.

## 1. Introduction

Traditional Chinese medicine (TCM) is attracting increasing attention globally due to its role in preventing and treating disease. With the ongoing investigations of the active ingredients and mechanisms of action, a growing number of TCM prescriptions are showing good therapeutic effects against many diseases, such as Alzheimer’s disease (AD), diabetes, and COVID-19 [1,2,3]. In clinical practice, multiple TCM decoction slices are used to form the compound formulae needed to conform to the compatibility principles, aiming to improve the curative effects or reduce the toxic side effects [4]. Herbs can present as complex chemical systems, and it is assumed that the use of multiple TCM species can lead to much more complicated components in a given multiherb formula, rendering quality evaluations more challenging [5,6]. For the majority of the TCM preparations that have been utilized in the clinic, little knowledge is available with respect to their chemical compositions, which not only limits the elucidation of the pharmacological effects, but also hinders the establishment of their quality standards to ensure their correct use in the clinic [7]. Therefore, it is crucial to develop potent analytical techniques to unveil the chemical compositions (via multicomponent characterization) and assess the quality (multicomponent quantitation) of TCM compound preparations.

The ongoing advancements in instrumentation and software have greatly driven the elucidation of TCM compound preparations, helping to make the development process systematic, personalizable, and intelligent, with high levels of integrity [8,9,10]. For instance, the application of high-performance liquid chromatography–mass spectrometry (HPLC/MS) online can separate and identify dozens of components with high sensitivity, which has become one of the most commonly used tools in the multicomponent characterization of herbal components [11,12,13]. In particular, versatile modern mass spectrometers can enable the use of alternative MS scan methods for different experimental purposes, which are mainly divided into data-independent acquisition (DIA) and data-dependent acquisition (DDA) techniques [14]. DIA approaches enable the full-coverage acquisition of all precursor ions, which typically include MS^E^, high-definition MS^E^ (HDMS^E^), sequential-window acquisition for all theoretical fragments (SWATH), and all-ion fragmentation (AIF) [15]. Because the fragments of all precursor ions recorded in the MS^1^ via fragmentation or sequentially windowed acquisitions are obtained, there is a need to match the primary precursor product ions prior to the data interpretation process, which is often conducted using commercial software or in-house algorithms [16,17]. DDA approaches are able to automatically target the most intense ions screened by full-scan MS^1^ to trigger the MS/MS or MS^n^ fragmentation, and the obtained spectra are easily analyzed due to the definite relationships between the precursor and product fragments [18]. However, when facing a complex matrix, DDA techniques often have low coverage for the components of interest. This issue can be partially solved by adding the precursor ions list (PIL) containing the target masses or the exclusion list (EL) containing the meaningless masses [19,20]. It is very impressive that ion mobility mass spectrometry (IM-MS) has been increasingly applied in TCM analysis recently, which can enable the additional separation of the gas-phase ions according to their charge, size, and shape, leading to more confident characterization of the chemical components of TCM compounds [21]. When IM-MS is combined with UHPLC, the four-dimensional information for each compound (*t*_R_, drift time, MS^1^, and MS^2^) can be obtained, which also shows great potential in distinguishing isomers [22,23].

As the first antiarrhythmic Chinese herbal drug approved by China Food and Drug Administration (CFDA), Wenxin granule (WXG), a five-herb TCM formula consisting of Codonopsis Radix (*Codonopsis pilosula* (Franch.) Nannf.; **CP**), Polygonati Rhizoma *(Polygonatum sibiricum* Red.; **PS**), Notoginseng Radix et Rhizoma (*Panax notoginseng* (Burk.) F. H. Chen; **PN**), Ambrum (Resin of Pinaceae), and Nardostachyos Radix et Rhizoma (*Nardostachys jatamansi* DC.; **NJ**), is clinically used to treat cardiac diseases such as arrhythmia and heart failure [24]. Previous studies have demonstrated that WXG exhibited its cardioprotective roles by inhibiting the inflammatory reaction, decreasing oxidative stress, regulating vasomotor disorders, and lowering cell apoptosis, and could play a role in treating cardiac arrhythmia through a complex multichannel inhibition process, involving inhibition of the transient potassium outward current (I(to)), late sodium current (I(NaL)), L-type calcium current (I(CaL)), and others [25,26,27]. Additionally, ginsenoside Rg1, ginsenoside Re, notoginsenoside R1, and lobetyolin were found to be the active compounds of WXG, which contributed to its cardioprotective effects [28]. Due to its good clinical efficacy, the comprehensive research on the base materials of WXG is necessary, which will also be the basis for studying its mechanism of intervention in complex diseases. A UHPLC/QTOF-MS approach was developed to analyze the main components of WXG, and 68 chemical components were identified [29], while an HS-SPME-GC-MS method identified 52 volatile components [30]. Moreover, an HPLC-UV method at 210 nm simultaneously determined the contents of four ginsenosides (noto-R1, Rg1, Rb1, and Rd) and lobetyolin in six batches of WXG samples [31]. In addition to these reports, further quality research on WXG is needed to establish more potent analytical techniques to target more components.

In this study, we integrated both qualitative (multicomponent characterization) and quantitative assays (multicomponent quantitation) to evaluate the quality of WXG, with the overall technical route illustrated in Figure 1. In detail, the qualitative analysis of WXG, the aim of which was to deeply characterize the multicomponents, was achieved via a high-definition MS^E^–high-definition DDA (HDMS^E^-HDDDA) hybrid scan approach using a Vion^TM^ IM-QTOF mass spectrometer coupled to a reversed-phase UHPLC system [22,32]. The quantitative assays of the 24 potential quality markers were performed via scheduled multiple reaction monitoring (sMRM) using a QTrap 4500 mass spectrometer after UHPLC separation. To achieve better performance, the key parameters affecting the chromatographic separation during UHPLC (involving the stationary phase, column temperature, and gradient elution program) and the detection conditions for both QTOF-MS (e.g., capillary voltage, cone voltage, and collision energy for HDMS^E^ and HDDDA) and QTrap-MS (sMRM-related parameters) were optimized. To facilitate the highly efficient and more reproducible characterization of WXG components, an intelligent workflow was established using UNIFI^TM^ software to process the obtained HDMS^E^ and HDDDA data. Up to 71 reference compounds (Figure 2; Appendix A) were used to aid the MS-oriented multicomponent characterization of WXG, and the quality of 27 batches of WXG samples (Appendix A) was assessed in this work. Hopefully, this can be an example of a comprehensive quality evaluation of TCM formulae.

## 2. Results and Discussion

### 2.1. Optimization of the RP-UHPLC/IM-QTOF-MS Approach Enabling the Profiling and Characterization of the Multicomponents from WXG

A dimension-enhanced LC-MS approach was developed to characterize the multicomponents from WXG using the Vion IM-QTOF-MS platform. In the first step, certain key parameters that can affect the chromatographic separation and MS detection were optimized via single-factor experiments.

#### 2.1.1. RP-UHPLC Conditions

Stationary-phase screening was performed by assessing 20 commercial reversed-phase (RP) columns with different silica gel cores and bonding technologies (from Agilent, Waters, Phenomenex, and Exmere Ltd.), with their information detailed in Appendix A. In this regard, the number of resolved peaks (MS-resolved peaks after UHPLC separation, obtained by applying UNIFI to process the negative-mode MS^1^ data of sample 2008021) and peak shape were taken into account as the indicators. The comparison results for the representative stationary phase (including HSS T3, Zorbax SB-C18, BEH C18, and HSS C18 SB) are shown in Figure 3, and detailed comparison results for 20 chromatographic columns are shown in Appendix A. Notably, to grasp the general differentiation points, a unified, unoptimized gradient elution program was utilized and each column was controlled at 35 °C. In general, the chromatographic resolution and peak shape on the HSS T3 column were more satisfactory, despite some other columns, such as ZORBAX SB-C18 (4408), BEH C18 (3325), and HSS C18 SB (2765), resolving more peaks. Additionally, by observing the BPI chromatograms, HSS T3 enabled a relatively good response to the majority of peaks. Furthermore, the universal silica-based bonded phase used for the HSS T3 sorbent is compatible with the 100% aqueous mobile phase and can enhance the retention of polar molecules, which is expected to result in better separation for the polar components of WXG, such as the ginsenosides and steroid saponins [29,31]. Accordingly, the HSS T3 column was selected. Subsequently, column temperatures ranging from 25 °C to 40 °C were used to observe the impacts on the resolution for WXG ingredients (Appendix A). Notably, with the increase in temperature, the resolution of the chromatographic peaks in the range of 22–25 min increased and the peak shape improved. Therefore, the column temperature of the HSS T3 column was set to 40 °C. To obtain a stable baseline and a more balanced chromatographic peak distribution, the gradient elution program was further adjusted and the optimal chromatography performance was acquired within 54 min.

#### 2.1.2. IM-QTOF-MS Conditions

The ion source parameters (capillary voltage and cone voltage) determine the overall ion response, while the collision energy applied in the collision cell (ramp collision energy (RCE) for HDMS^E^; mass-dependent RCE (MDRCE) for HDDDA) affects the generation of fragments that are useful for the structural elucidation process. In this work, these parameters were optimized using a mass spectrometer operating in both negative and positive ESI modes. Given the component diversity of WXG, eight representative components, including ginsenosides Re, -Rb1, -Ro, -F2, notoginsenoside R1 (noto-R1), chlorogenic acid, tangshenoside, and lobetyolin, were selected for the index in the negative mode (ESI−), while codonopsinol A, hypoxanthine, codotubulosine B, acacetin, trillin, and quercetin were considered in positive mode (ESI+). For ESI−, the variation tendencies of the ion responses for these components among the different capillary voltage (1.0–3.5 kV) and cone voltage (20–100 V) settings are shown by the histograms in Figure 4. By comparing the peak areas in the extracted ion chromatogram (EIC), it was found that with the increase in capillary voltage, the responses of chlorogenic acid, tangshenoside I, Ro, and noto-R1 basically decreased, while the other components (lobetyolin, Re, Rb1, and F2) first became more abundant and then weaker. Therefore, the capillary voltage at 1.5 kV in negative mode was selected. Due to the ion responses of the six indicators decreasing with the increase in capillary voltage, 1 kV was chosen in positive ion mode. Five groups of cone voltages were optimized in both positive and negative ion modes: 20 V, 40 V, 60 V, 80 V, and 100 V. The results indicated that when the cone voltage exceeded 60 V, the peak area values of chlorogenic acid, tangshenoside I, lobetyolin, noto-R1, Re, and Ro showed a decreasing trend in negative mode, while the peak areas of Rb1 and F2 increased. The results also proved that in positive ion mode, when the cone voltage was 20 V, the peak areas of the four components were the largest (expect for quercetin and acacetin). Consequently, the cone voltage was set to 60 V in negative ion mode and 20 V in positive ion mode. The Vion IM-QTOF hybrid high-resolution mass spectrometer can enable the RCE in HDMS^E^ and the MDRCE in HDDDA. For RCE in HDMS^E^, in the negative ion mode, three energy groups at 10–50, 20–60, and 30–70 eV were compared, and 20–60 eV was selected by considering the richness of the MS^2^ fragments of ginsenosides, flavonoids, organic acids, and the phenylpropanoid. The four energy groups at 10–40, 20–50, 30–60, and 15–40 eV were compared in positive ion mode, and 15–40 eV was selected by considering the abundance of the secondary fragments of ginsenosides, alkaloids, steroidal saponins, and the other components. Differing from the RCE, the MDRCE in DDA mode can enable a targeted collision energy ramp on the precursors through the predefined scan range. According to the same selection rules as used for comparing the RCE in HDMS^E^, four energy groups (10–30/40–60, 20–40/50–70, 30–50/70–90, and 15–30/30–40 eV) in negative ion mode and five energy groups (10–20/20–30, 20–30/30–40, 30–40/40–50, 40–50/50–60, 15–20/20–35 eV) in positive mode were compared for MDRCE values in HDDDA. We finally selected MDRCE values of 15–30/30–40 eV in the negative ion mode and 15–20/20–35 eV in the positive ion mode to obtain the diversified product ions for most of the compounds. Detailed information is displayed in Appendix A.

The Vion IM-QTOF analytical platform can facilitate several DIA and DDA scan techniques, as well as allowing ion mobility separation. The HDMS^E^-HDDDA hybrid scan method is a relatively novel scanning strategy, which can enable the ion mobility separation of all precursor ions generating the high-definition MS^1^ spectrum, while the alternate DIA and DDA acquisitions can record the HDMS^E^ and HDDDA spectra with CID-MS^2^ (collision-induced dissociation) data [20,21,22,32]. In this work, targeting multiple classes of components from WXG, we established the hybrid HDMS^E^-HDDDA approaches in both the negative and positive modes, and in each mode one scan cycle could generate an IM-enabled full-scan MS^1^, an MS^2^ spectrum of HDMS^E^, and two MS^2^ spectra of HDDDA (top 2). Automated annotation of the obtained HDMS^E^ and HDDDA data was achieved by applying the UNIFI solution, which had been incorporated in an in-house chemical library. The in-house library contained 1129 known compounds, of which 556 from PN, 174 were from CP, 135 were from PS, and 264 were from NJ. Additionally, by enabling the IM function, the CCS information for the components of WXG could be acquired, which could also separate the co-eluting and isomeric components (Appendix A). For instance, additional peaks were resolved at retention times of 13.79, 18.38, 27.21, 36.44, 41.88, and 46.14 min, and for isomers with *m*/*z* values of 357.13, 469.13, 683.43, 815.47, 1077.51, and 1239.63. These cases demonstrate the superiority of IM separation in the profiling and characterization of the multicomponents from WXG.

### 2.2. Comprehensive Characterization of the Chemical Components of WXG by Analyzing the HDMS^E^-HDDDA Data Using the UNIFI Workflows

Comprehensive characterization of the multicomponents of WXG was accomplished by analyzing the high-resolution CID-MS^2^ data obtained via HDMS^E^-HDDDA hybrid scans in both negative and positive ion modes. By utilizing the intelligent UNIFI workflows, automatic peak annotation was achieved by searching the in-house chemical library. The HDDDA data were first analyzed for structural elucidation because of the high quality of the MS^2^ spectra, while the HDMS^E^ data were comparatively analyzed as a supplement to the characterization results. By comparing these results with the reference compounds (Figure 2), analyzing the fragmentation information, searching the in-house library, and referring to the relevant literature, a total of 205 components were identified or tentatively characterized from WXG, including 28 components identified with the aid of reference compounds. Figure 5 shows the base peak chromatograms (BPCs) of WXG in both negative and positive ion modes, with all characterized peaks annotated. Detailed information with respect to these components is given in Appendix A.

#### 2.2.1. Characterization of Ginsenosides

Ginsenosides are the main active ingredients in WXG, originating from PN (*Panax notoginseng*) [28,33]. We were able to characterize 92 ginsenosides (including 26 ginsenosides confirmatively identified with the aid of reference compounds) in the current work, whose structures featured a triterpene sapogenin and one or two sugar chains. The characterization of ginsenosides from WXG mainly relied on the negative CID-MS^2^ data, as severe in-source decay occurred when the important structure information was missed, as in our previous report [34]. Based on the current knowledge of the chemistry of *Panax* and the saponin reference compounds [8], the sugars of ginsenosides included Glc (consistent with the typical neutral loss (NL) of 162.05 Da), GlurA (176.03 Da), Rha (146.06 Da), and Xyl or Ara (132.04 Da) [35]. Additionally, the diagnostic product ion (DPI) forms, involving *m*/*z* 475.38 for protopanaxadiol (PPD), *m*/*z* 459.38 for protopanaxatriol (PPT), *m*/*z* 455.35 for oleanolic acid (OA), and *m*/*z* 491.37 for octillol (OT), could be used to rapidly and primarily identify the subtypes of ginsenosides [36]. For convenient expression, Xyl was used to represent the pentose featured by NL of 132 Da in the characterization of ginsenosides.

Compound #**135** (*t*_R_ 29.11 min) was identified as notoginsenoside R4 (C_59_H_100_O_27_) by comparison with the reference standard, and its fragmentation features were useful to characterize those unknown PPD-type saponins. Its deprotonated precursor ion ([M−H]^–^) was observed at *m*/*z* 1239.6390. The subsequential cleavages of Xyl and two Glc residues generated the main fragment ions at *m*/*z* 1107.5948, 945.5413, and 783.4900. On this basis, another two Glc residues were eliminated to yield the fragment of *m*/*z* 459.3826, which was the characteristic fragment ascribed to the deprotonated PPD sapogenin (Appendix A). In a similar manner, an unknown compound #**184** (*t*_R_ 41.89 min; [M−H]^–^ at *m*/*z* 945.5432) was identified as PPD-3Glc. Based on its fragment information that was analogous to ginsenoside Rd (consistent with compound #**180**, *t*_R_ 40.52 min), compound #**184** was tentatively characterized as gypenoside XVII or its isomer by comparing it with the database and the literature [37]. Compound #**72** (*t*_R_ 20.39 min) was identified as notoginsenoside R1 (C_47_H_80_O_18_), a typical PPT-type ginsenoside, by comparison with the reference standard. Its formic acid adduct ion was observed at *m*/*z* 977.5326, which, upon CID-MS/MS fragmentation, yielded the product ions at *m*/*z* 799.4834, 637.4312, and 475.3772, ascribed to the neutral elimination of Xyl, Xyl+Glc, and Xyl+2Glc, respectively. The sapogenin ion of *m*/*z* 475.3772 was consistent with the PPT moiety (Appendix A). These fragmentation behaviors were very similar to those observed for the PPD-type ones. In the case of an unknown saponin, compound #**86** (*t*_R_ 22.88 min) gave the [M−H]^–^ ion at *m*/*z* 901.5166 (C_46_H_78_O_17_). The CID-MS/MS cleavage generated the fragments at *m*/*z* 769.4731 ([M–H–Xyl]^–^), 607.4235 ([M–H–Xyl–Glc]^–^), and 475.3754 ([PPT−H]^–^), which indicated that two Xyl residues and one Glc residue attached to the PPT sapogenin. After searching the in-house library, we tentatively identified compound #**86** as chikusetsusaponin LM2 or its isomer [38].

Some rare OA- and OT-type ginsenosides were tentatively characterized from WXG. OA ginsenosides were often glycosylated at 3-OH and 28-COOH, and gave the characteristic sapogenin fragment *m*/*z* 455.35 [21]. Compound #**197** (*t*_R_ 45.20 min) gave the deprotonated molecular ion peak at *m*/*z* 925.4806 (C_47_H_74_O_18_). The secondary fragment of *m*/*z* 731.4372 should result from the neutral elimination of GlcA+H_2_O from the precursor ion, which could further lose another Glc generating the fragment of *m*/*z* 569.3852 (Figure 7). The deprotonated OA fragment was detected at *m*/*z* 455.3515 by eliminating all attached sugars (GlcA, Glc, and Xyl). By searching the in-house library, compound #**197** was tentatively identified as OA-GlurA-Glc-Xyl, plausibly being 3-*O*-*β*-d-xylopyranosyl-(1→2)-*β*-d-glucopyranosyl-28-*O*-*β*-d-glucopyranosyl [39]. For the OT ginsenosides, a unique sugar chain was glycosylated at 6-OH and the sapogenin ion was typically at *m*/*z* 491.37. Compound #**66** (*t*_R_ 19.49 min) was an OT-type ginsenoside, which gave the formic acid adduct precursor ion at *m*/*z* 831.4738 (C_41_H_70_O_14_). After the sequential elimination of Xyl and Glc, the product ions of *m*/*z* 653.4335 and 491.3745 were acquired, and accordingly this compound was identified as OT-Glc-Xyl, which was presumed to be 24(*R*)-pseudoginsenoside Rt2 or its isomer by comparing it with the database (Figure 7).

#### 2.2.2. Characterization of Steroidal Saponins

The steroidal saponins in WXG were from PS (*Polygonatum sibiricum*) [40], which showed rich fragmentation in the positive mode. A total of 52 steroidal saponins were identified or tentatively characterized from WXG in this work. Differing from ginsenosides, the precursor information was obtained in the protonated form ([M+H]^+^). Compound #**162** (*t*_R_ 36.36 min) exhibited the protonated precursor ion at *m*/*z* 885.4815, suggesting the molecular formula of C_45_H_72_O_17_. Upon the CID-MS/MS fragmentation, the characteristic fragments ions at *m*/*z* 723.4322, 577.3721, and 415.3714 were observed, corresponding to the sequential elimination of Glc, Rha, and Glc, respectively (Figure 6). With the aid of the database and the literature, this was tentatively identified as gracillin or its isomer [41]. The molecular ion peaks ([M+H]^+^) of compound #**56** (*t*_R_ 18.73 min) and #**62** (*t*_R_ 18.89 min) were generated at *m*/*z* 915.4604 and 1047.5086, respectively, with a series of similar characteristic fragments at *m*/*z* 753.4087 and 591.3524 and the same aglycone ion at *m*/*z* 429.2997, presumably with two Glc residues connected on a steroidal sapogenin. According to the discrepant fragments at *m*/*z* 915.4616 and 1047.5086, compound #**56** exhibited NL of 162 Da from *m*/*z* 915 to 753, while compound #**62** lost Gal and Xyl successively based on the transition from *m*/*z* 1047 to 753. They were finally speculated to be (25*S*)-pratioside D1 or its isomer and spirost-5-en-12-one-3-O-*β*-d-glucopyranosyl-(1→2)-[*β*-d-xylopyranosyl-(1→3)]-*β*-d-glucopyranosyl-(1→4)-*β*-d-galactopyranoside or its isomer [42,43].

#### 2.2.3. Characterization of Alkaloids

A total of 14 alkaloids were characterized from WXG, which showed good responses in the positive mode, and 10 alkaloids were then confirmed from CP (*Codonopsis pilosula*) [44]. Compound #**27** (*t*_R_ 14.10 min) gave the precursor ion of [M]^+^ at *m*/*z* 350.1955, suggesting the molecular formula to be C_19_H_28_NO_5_^+^ (Figure 6). By analyzing the MS/MS spectrum, the product ion at *m*/*z* 250.1432 was acquired via the loss of one H_2_O and one C_5_H_6_O. Because of the cleavage of C_4_H_11_NO and C_2_O, the fragment ions at *m*/*z* 161.0583 and 121.0632 were generated. Combined with searching the database and literature, this compound was tentatively assigned to be codonopyrrolidium A or its isomer [45]. Compared with the literature [29], here alkaloids were newly characterized from WXG.

#### 2.2.4. Characterization of Organic Acids

Organic acids, including 9 compounds, were characterized from WXG. The characteristic NL of CO, CO_2_, and H_2_O was readily observed in the negative CID-MS^2^ images of organic acids. For instance, compound #**40** (*t*_R_ 17.99 min) showed abundant levels of the precursor ion of [M−H]^–^ at *m*/*z* 515.1195, corresponding to the molecular formula of C_25_H_24_O_12_. The fragment at *m*/*z* 353.0865 was generated from the precursor ion due to the cracking of C_9_H_6_O_3_ (162.03 Da), plausibly being a caffeoyl substituent. Another product ion of *m*/*z* 135.0432 could be further evidence supporting the presence of caffeoyl. The fragment at *m*/*z* 179.0323 could be deprotonated quinic acid, which was further dissociated into the ion of *m*/*z* 173.0448 by losing H_2_O (Figure 7). Accordingly, in combination with the database retrieval, compound #**40** was tentatively identified as di-*O*-caffeoylquinic acid.

In addition to these classes of components, we also characterized 7 flavonoids, 8 terpenoids, and 22 other compounds from WXG. The typical cases using compound #**24** (*t*_R_ 13.80 min) for terpenoids and #**79** (*t*_R_ 22.17 min) for flavonoids are illustrated in Figure 7. Compared with the previous study [29], through the optimization of chromatographic and MS conditions and using the HDMS^E^-HDDDA hybrid scan method, we identified many more chemical components from WXG (205 VS 68) and increased the characterization of alkaloids.

### 2.3. Development and Validation of the UHPLC-sMRM Approach for Quantifying 24 Components from WXG

To evaluate the quality of multibatch commercial WXG samples, a UHPLC-sMRM approach was established and validated. A total of 24 compounds, including three from CP (lobetyolin, syringin, and atractylenolide III), and 21 from PN (noto-R1, noto-R2, noto-Fa, Re, Rg1, Rf, Rg2, Rh1, Rb1, Ra1, Rb2, F1, chikusetsusaponin IV, Rd, F4, Rk3, F2, Rh4, Rg3, Rk1, and Rg5) were quantitatively assayed. Lobetyolin, syringin, and atractylenolide III were mainly used for the quality evaluation of CP [46,47]. Oxidative stress and inflammatory reactions are closely related to cardiovascular and cerebrovascular diseases. *Panax notoginseng* saponins (PNS) were the main active ingredients of PN, which could play an important role in ameliorating inflammation, attenuating oxidative damage, and promoting angiogenesis in a myocardial ischemia model [48,49,50].

#### 2.3.1. Method Development

We examined the extraction conditions as the first step in the development of the UPHLC-sMRM approach. In this step, the effects of using water, acetonitrile, and methanol as the extraction solvents were evaluated. As evidenced in Appendix A, increasing the ratio of water could reduce the extraction rate of the hydrophobic components (Rg3, Rk1, and Rg5), and a higher extraction rate was obtained in methanol. Subsequently, we further compared the extraction times via ultrasonic extraction with methanol. The third extraction cycle led to a low yield compared to the first cycle; therefore, two cycles of ultrasonic extraction using methanol were used for sample preparation in the quantitative assay experiment. To develop the sMRM method, key parameters (including CE, DP, Q1 mass, and Q3 mass) for 24 target components and IS were optimized in ESI+ and ESI– modes to screen out abundant and interference-free precursor-to-product ion transitions (Table 1). The optimization process was achieved by perfusing 24 standard solutions (200 ng/mL) at a constant flow of 7 μL/min using a needle pump only through the mass spectrometer in Q1 scan, product ion scan, and MRM modes. Notably, astragaloside IV was selected as the internal standard because it was not detected in WXG, CP, PN, or PS, and showed good resolution with the other analytes. Furthermore, the structure of astragaloside IV was similar to the ginsenosides, and its chemical properties were stable. We found that except for noto-Fa, chikusetsusaponin IV, and atractylenolide III, the remaining 21 components were prone to generating the precursor ions of [M+HCOO]^–^ with formic acid as the additive in the water phase. The ginsenoside isomers, such as Rk3, Rh4, F2, and Rg3, were fragmented, yielding almost the same product ions, and accordingly the same ion pair was selected for monitoring. Under the current chromatographic conditions, isomeric analytes could be baseline-separated. The gradient elution program used for UHPLC/QTrap-MS was carefully optimized to enable rapid, sensitive, and accurate analysis. The acquisition window was set to ±30 s around the reference standard retention time (Figure 8). The algorithm from Analyst software automatically optimized the cycle time and dwell time to achieve a lower coefficient of variation and lower detection limit.

#### 2.3.2. Method Validation

The developed UHPLC-sMRM method was further validated in terms of its selectivity, linearity, sensitivity, precision, stability, repeatability, recovery, and matrix effect [51] (Appendix A). By comparing the sMRM chromatograms obtained by injecting methanol, blank methanol spiked with reference standards, and QC, the target analytes were free from interference of the complex matrix in the predefined acquisition window (Appendix A). The linear regression determination coefficient (*r*) values for 24 analytes were between 0.9956–0.9995 over the corresponding linear concentration ranges. The LODs of all analytes varied over the range of 0.0031–6.25 pg, and the LOQs varied between 0.0039 and 25 pg. The RSD% values of intra-day and inter-day precision were found to be lower than 14.0% for all concentration levels (low, medium, and high) of the QC samples. The recovery rates were between 85.6% and 108.3%. Stability rates within 72 h for all analytes were in the range of 3.2–7.7%, and the repeatability test showed variation rates of less than 9.9%. No remarkable matrix effect was observed, as all analytes gave slope ratio values of 85.1–109.0%. Above all, the established UHPLC-sMRM approach could act as a reliable method to quantitatively assay 24 target compounds from WXG.

### 2.4. Quantitative Evaluation of the Multibatch WXG Samples

The validated UHPLC-sMRM quantitative assay method was finally used to detect 27 batches of commercial WXG samples targeting 24 analytes, and the results are given in Appendix A and visualized by the histograms in Figure 9A. In general, the total content levels for these 24 compounds among 27 batches of WXG samples varied between 30.34 and 46.63 mg/bag. In particular, the sum amounts of Rg1, Rb1, and noto-R1, in all WXG samples were in the range of 22.80–36.47 mg/bag, which were higher than the content limit of 17.0 mg/bag (Figure 9C) required by the quality standards of the Chinese Pharmacopoeia (2020 edition). Batch 14 contained the highest amounts of Rg1, Rb1, and noto-R1, reaching 36.47 mg/bag. Moreover, when evaluated using ginsenosides, the sum content variations of 21 ginsenosides and 3 ginsenoside markers recorded by the Chinese Pharmacopoeia in 27 batches of WXG samples were very similar (Figure 9B,C). The 5 common notoginseng saponins, namely noto-R1, Rg1, Re, Rb1, and Rd, accounted for about 90% of the content out of the 21 ginsenosides (25.58–40.69 mg/bag). In addition, we obtained the content intervals for the sum values of lobetyolin, syringin, and atractylenolide III, which were in the range of 0.14–0.37 mg/bag (Figure 9D). The results reported in the current work demonstrate the potency of this UHPLC-sMRM approach in the quality control of WXG by comprehensively reflecting the content changes of PN and CP.

## 3. Materials and Methods

### 3.1. Chemicals and Materials

A total of 71 compounds (Figure 2 and Appendix A, purity ≥ 98.0% by HPLC) were used as the reference compounds, including 57 saponins, 2 organic acids, 5 flavonoids, and 7 others. They were purchased from Shanghai Yuanye Biotech. Co., Ltd. (Shanghai, China) or Chengdu Desite Biotech. Co., Ltd. (Chengdu, China). Acetonitrile, methanol (Fisher, Fair lawn, NJ, USA), and formic acid (ACS, Wilmington, DE, USA) were of HPLC grade. Ultra-pure water was in-house-prepared using a Milli-Q water purification system (Millipore, Bedford, MA, USA). In total, 27 batches of WXG samples (Appendix A) from Shandong Buchang Pharmaceutical Co., Ltd., were collected, which were divided into two different specifications depending on whether sucrose was contained (no. 1–14 with sucrose, 9 g/bag; no. 15–27 without sucrose, 5 g/bag). All specimens were deposited in the State Key Laboratory of Component-Based Chinese Medicine (Tianjin, China).

### 3.2. Sample Preparation of WXG

To prepare the WXG sample for the multicomponent characterization, the accurately weighed WXG sample was dissolved in 40% aqueous methanol (*v/v*) and further vortexed for 2 min. Extraction was performed ultrasonically (400 W, 40 kHz) at 25 °C for 1 h. Then, the liquid underwent centrifugation at 11,481× *g* (14,000 rpm) for 10 min, leading to the supernatant used as the test solution for WXG (concentration: 50 mg/mL). Meanwhile, the proper amounts of CP, PS, and PN samples were prepared as the positive control to identify the multicomponents of WXG. The powders of CP and PS were mixed with ultrapure water, while PN was mixed with 80% aqueous ethanol (*v/v*). These three herbs were extracted under reflux for 1 h, and the resultant liquid was further centrifuged at 11,481× *g* at 4 °C for 10 min to prepare the sample solutions (approximately 100 mg/mL).

To prepare the WXG sample for the quantitative assays, the following method was utilized. The precisely weighed WXG powder (100 mg) was added to 4 mL of methanol and mixed well. The mixture was ultrasonically extracted (400 W, 40 kHz) for 1 h. After being centrifuged at 3219 g (4000 rpm) at 4 °C for 10 min, the resulting supernatant was transferred into a 10 mL volumetric flask. The extraction process was repeated and the resultant supernatants were pooled and further diluted to a constant volume. The extract was filtered through a 0.22 μm nylon membrane. For the final WXG sample used for the quantitative assay experiment, 100 μL of the WXG solution, 30 μL of the internal standard solution (astragaloside IV 5 μg/mL), and 70 μL of methanol were mixed and centrifuged at 11,481 g for 10 min.

### 3.3. Preparation of the Reference Standard Solutions for Calibration Curves

Individual stock solutions of 25 compounds (24 analytes and IS) were prepared by dissolving the reference standards in methanol or a mixture of water and methanol (1:1, *v/v*). Then, all the stock solutions were combined at a certain proportion to yield the mixed reference standards solution, which was further diluted and generated a series of calibration solutions using methanol. The internal standard working solution was 100 μg/mL of astragaloside IV in methanol. One WXG sample, with the batch number of 2105016, served as the quality control (QC) sample.

### 3.4. UHPLC/IM-QTOF-MS

The high-accuracy MS data for the multicomponent profiling and characterization of WXG were acquired through the use of the ACQUITY UPLC I-Class/Vion^TM^ IM-QTOF system (Waters Corporation, Milford, MA, USA). An ACQUITY UPLC HSS T3 column (2.1 × 100 mm, 1.8 µm) maintained at 40 °C was selected for the chromatographic separation. A binary mobile phase, composed of solvent A (0.1% formic acid in water) and solvent B (acetonitrile), was employed at a flow rate of 0.3 mL/min following an optimal gradient program: 0–2 min, 2% (B); 2–5 min, 2–10% (B); 5–10 min, 10% (B); 10–11 min, 10–15% (B); 11–16 min, 15–20% (B); 16–22 min, 20–27% (B); 22–25 min, 27–30% (B); 25–34 min, 30% (B); 34–38.5 min, 30–36% (B); 38.5–52 min, 36–65% (B); and 52–54 min, 65–98% (B).

High-accuracy MS data were recorded on the Vion IM-QTOF mass spectrometer in both negative and positive modes (Waters Corporation, Milford, MA, USA). The LockSpray ion source was equipped using the following parameters: capillary voltage, 1.5 kV (ESI−), 1.0 kV (ESI+); cone voltage, 60 V (ESI−), 20 V (ESI+); source offset, 80 V; source temperature, 120 °C; desolvation gas temperature, 500 °C; desolvation gas flow (N_2_), 800 L/h; cone gas flow (N_2_), 50 L/h. Default parameters were defined for the travelling wave IM separation. The CID-MS^2^ data in both negative and positive modes were acquired using the HDMS^E^-MSMS hybrid scan approach. In detail, the QTOF analyzer scanned over a mass range of *m*/*z* 80–1500 at a low energy of 6 eV for both the negative and positive HDMS^E^ and HDDDA modes at 0.3 s per scan (MS^1^). Ramp collision energy (RCE) ranges of 15–40 eV and 20–60 eV were set in the positive and negative HDMS^E^ modes. For HDDDA settings, when the TIC (total ion chromatogram) intensity exceeded 200 detector counts, MS/MS fragmentation of the two most intense precursors was automatedly triggered, which stopped after no longer than 0.4 s (time-out). MDRCE ranges of 15–20 eV in the low mass ramp and 20–35 eV in the high mass ramp were set in the positive mode, while ranges of 15–30 eV in the low mass ramp and 30–40 eV in the high mass ramp were set in negative mode. MS data calibration was conducted by constantly infusing the leucine enkephalin solution (Sigma-Aldrich, St. Louis, MO, USA; 200 ng/mL) at a flow rate of 10 µL/min. The calibration of CCS was conducted according to the manufacturer’s guidelines using a mixture of calibrants.

### 3.5. Data Processing

The uncorrected HDMS^E^ and HDDDA data for WXG were further processed using UNIFI 1.9.3.0 (Waters), which could perform the data correction, peak picking, and peak annotation processes efficiently. The key parameters set in UNIFI were as follows. Find 4D peaks (only set in HDMS^E^): high-energy intensity threshold, 50.0 counts; low-energy intensity threshold, 100.0 counts. Find DDA masses (only set in HDDDA): MS ion intensity threshold, 100.0 counts; MSMS ion intensity threshold, 50.0 counts. Target by mass: target match tolerance, 10.0 ppm; screen on all isotopes in a candidate and generate predicted fragments from structure were enabled; fragment match tolerance, 10.0 ppm. Adducts: negative adducts including +HCOO, –H, +CH_3_COO, +Cl. Lock Mass: combine width, 3 scans; mass window, 0.5 *m*/*z*; reference mass, 554.2620; reference charge, −1. Positive adducts including +H, +Na, +K, +NH_4_, +Li. Lock Mass: combine width, 3 scans; mass window, 0.5 *m*/*z*; reference mass, 556.2766; reference charge, +1.

### 3.6. UHPLC/QTrap-MS Using Scheduled MRM (sMRM)

Multicomponent quantitative assays of 24 compounds in WXG were performed on the Waters ACQIUTY UPLC I-Class system (Waters, Milford, MA, USA) coupled with a QTrap 4500 mass spectrometer (AB Sciex Scientific, Concord, ON, Canada). The chromatographic separation conditions, including the chromatographic column, column temperature, mobile phase, and flow rate, were consistent with those depicted for the multicomponent characterization methods. The gradient elution program was different, representing a more efficient analytical approach: 0–1 min, 5–16% (B); 1–1.5 min, 16–23% (B); 1.5–4 min, 23–28% (B); 4–5 min, 28–30% (B); 5–10 min, 30–31% (B); 10–14 min, 31–33% (B); 14–21 min, 33–65% (B); and 21–23 min, 65–95% (B). The injection volume was 2 μL for each run. Regarding the ion source parameters, curtain gas and ion source gas 1 and gas 2 were maintained at 35, 55, and 55 psi, respectively; the collision gas was medium; the source temperature was set to 550 °C; ion spray voltages were fixed at 4500 V and −4500 V for the positive and negative modes, respectively. The key parameters, including the parent ion, product ion, collision energy (CE), declustering potential (DP), retention times of 24 analytes, and the internal standard (IS) were summarized in Table 1. The detection window of each ion pair was set to 1 min (actual acquisition window = RT ± 1/2 detection window). All raw data collected using the UHPLC-sMRM approach were processed using Analyst 1.7 software (AB Sciex Scientific, Vaughan, ON, Canada) for peak recognition and peak integration.

### 3.7. Method Validation of the UHPLC-sMRM Approach

The UHPLC-sMRM approach was validated in terms of the selectivity, linearity, sensitivity, precision, stability, repeatability, recovery, and matrix effect. In detail, the stock solution was serially diluted to obtain 8 calibration solutions with different concentrations. Calibration curves were constructed by establishing a linear regression function after weighting (1/x^2^) the relationship between the ratio of peak area analytes and IS against their corresponding concentrations. Under current chromatographic conditions, the lower limit of detection (LLOD) was assessed using an S/N ratio of 3, and the lower limit of quantification (LLOQ) corresponded to the S/N ratio of 10. The sMRM diagrams of the blank sample, spiked sample, and QC were examined to evaluate the selectivity. For intra-day and inter-day precisions expressed in RSD (%), three levels of the spiked analyte solution (low, mid, and high) were determined in six replicates within a day to assess the intra-day precision, and then the test was repeated for three consecutive days serving as the inter-day precision. The repeatability was estimated by analyzing six independent WXG samples in parallel, which were prepared using the optimized extraction method. The stability of the analyte was tested using QC samples sealed in an automatic sampler for 0 h, 3 h, 6 h, 9 h, 12 h, 24 h, 36 h, 48 h, 60 h, and 72 h. The accuracy of all verification tests was determined based on the recovery of each analyte. A recovery experiment was carried out by quantitatively adding mixed standards (low, medium, and high) to the WXG extract, and the calculation formula was as follows:(1)$$Recovery\;\left(\%\right)=\frac{\left(detected\;amount-original\;amount\right)}{addition\;}\times 100\%$$

Matrix-matched calibration curves were created using WXG samples added to a known amount of each analyst at seven spiking levels. The effect of the WXG matrix on the ionization efficiency was assessed by comparing the slopes of the matrix-matched standard curves (B) and the slopes of the standard solution curves (A). The matrix effect was calculated as follows:(2)$$Matrix\;effect\;\left(\%\right)=\frac{B}{A}\times 100\%$$

## 4. Conclusions

Aimed at the comprehensive multicomponent characterization and quality evaluation of commercial WXG, two LC-MS approaches (UHPLC/IM-QTOF-HDMS^E^-HDDDA and UHPLC-sMRM) were developed. In particular, the application of a hybrid HDMS^E^-HDDDA scan method, which enabled the IM separation of all precursor ions and the alternative DIA and DDA acquisition methods via once injection analysis, showed the advantages of obtaining high-definition MS^1^ and MS^2^ spectra with high data coverage while profiling and characterizing WXG complex components. The established intelligent data interpretation workflows achieved using UNIFI characterized 205 components (including 92 ginsenosides, 53 steroidal saponins, 14 alkaloids, 9 organic acids, 7 flavonoids, 8 terpenoids, and 22 others), which indicated the complexity of the non-volatile chemical components of WXG. Up to 24 compounds, including 21 ginsenosides from PN and 3 compounds from CP, were quantitatively assayed using a validated UHPLC-sMRM method, which were established on the QTrap 4500 mass spectrometer. The quality differences between 27 batches of commercial WXG samples were unveiled. The combination of multicomponent characterization and multicomponent quantitation processes is a practical solution for the evaluation of TCM compound preparations.

## Figures and Tables

**Figure 1 molecules-27-03647-f001:**
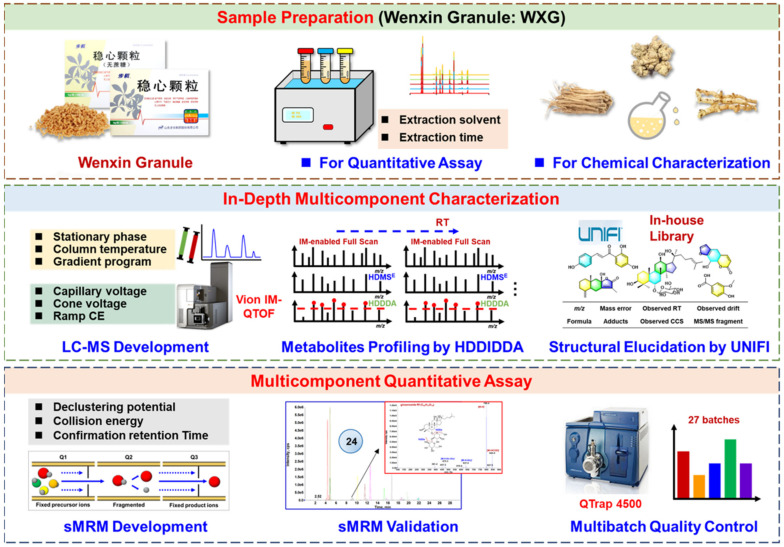
Quality evaluation of Wenxin granule (WXG) via UHPLC/IM-QTOF-MS-based multicomponent characterization and quantitative assay using scheduled MRM facilitated by UHPLC/QTrap-MS.

**Figure 2 molecules-27-03647-f002:**
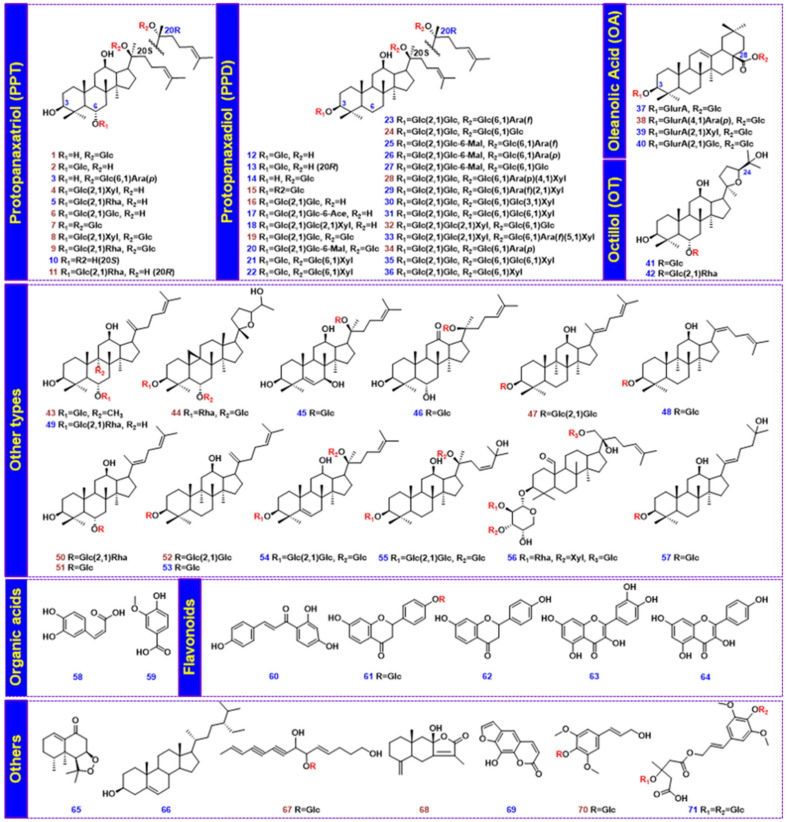
Chemical structures of the 71 reference compounds (quantitative markers are marked with the numbers in red).

**Figure 3 molecules-27-03647-f003:**
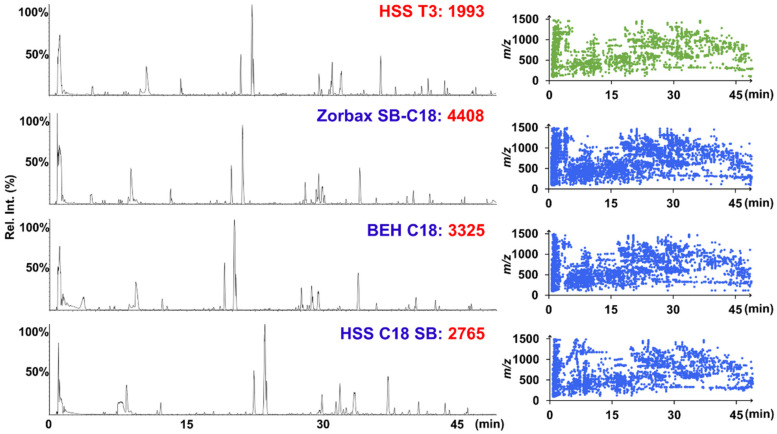
The representative UHPLC columns (including HSS T3, Zorbax SB-C18, BEH C18, and HSS C18 SB) used for the stationary-phase screening. The left shows the base peak intensity (BPI) chromatograms obtained on the candidate columns, while the right shows a scatter plot of the components resolved by both MS and chromatographic separation.

**Figure 4 molecules-27-03647-f004:**
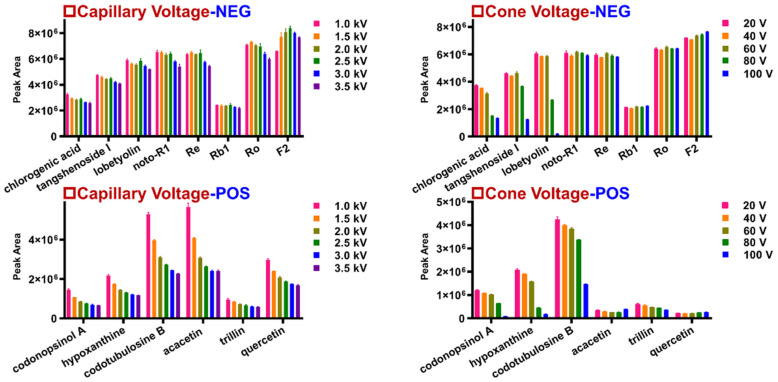
Optimization of the two key parameters (capillary voltage and cone voltage) in both the positive and negative ESI modes of the Vion IM-QTOF-MS system (*n* = 3).

**Figure 5 molecules-27-03647-f005:**
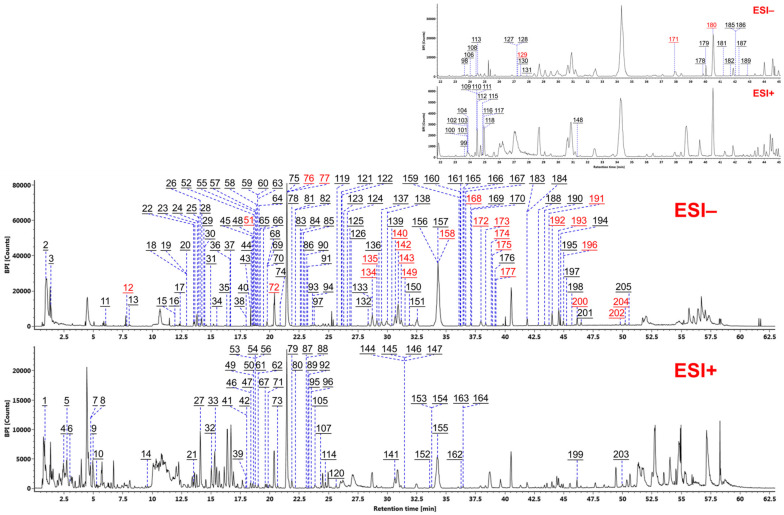
Base peak intensity (BPI) chromatograms of WXG in both positive and negative ESI modes. The peaks characterized with the aid of reference compounds are annotated in red. Peaks **98**–**104**, **106**, **108**–**112**, **115**–**118**, **127**–**131**, **148**, **171**, **178**–**182**, **185**–**187**, and **189** are marked separately in the upper right corner for better observation.

**Figure 6 molecules-27-03647-f006:**
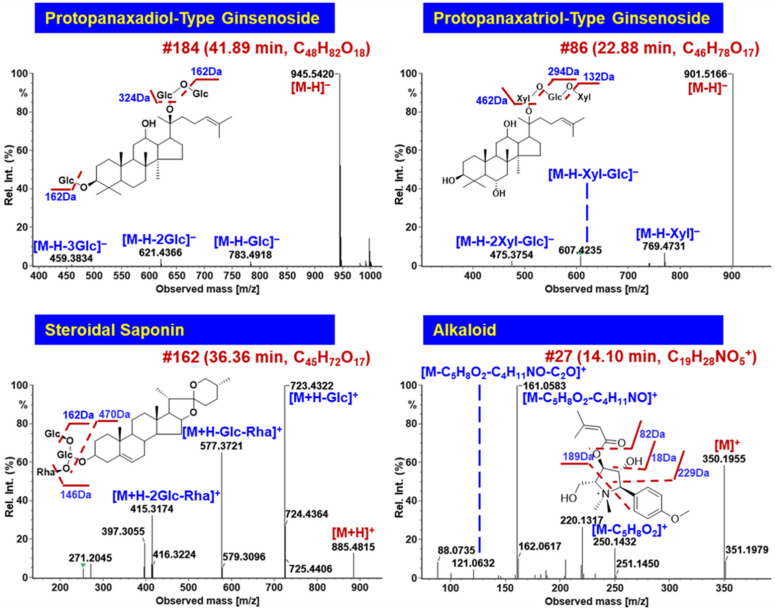
Annotation of the CID-MS^2^ spectra of the protopanaxadiol (PPD) and protopanaxatriol (PPT)-type ginsenosides, steroidal saponins, and alkaloids from WXG using UNIFI^TM^, showing the characteristic fragmentations and their application to characterize the unknown components.

**Figure 7 molecules-27-03647-f007:**
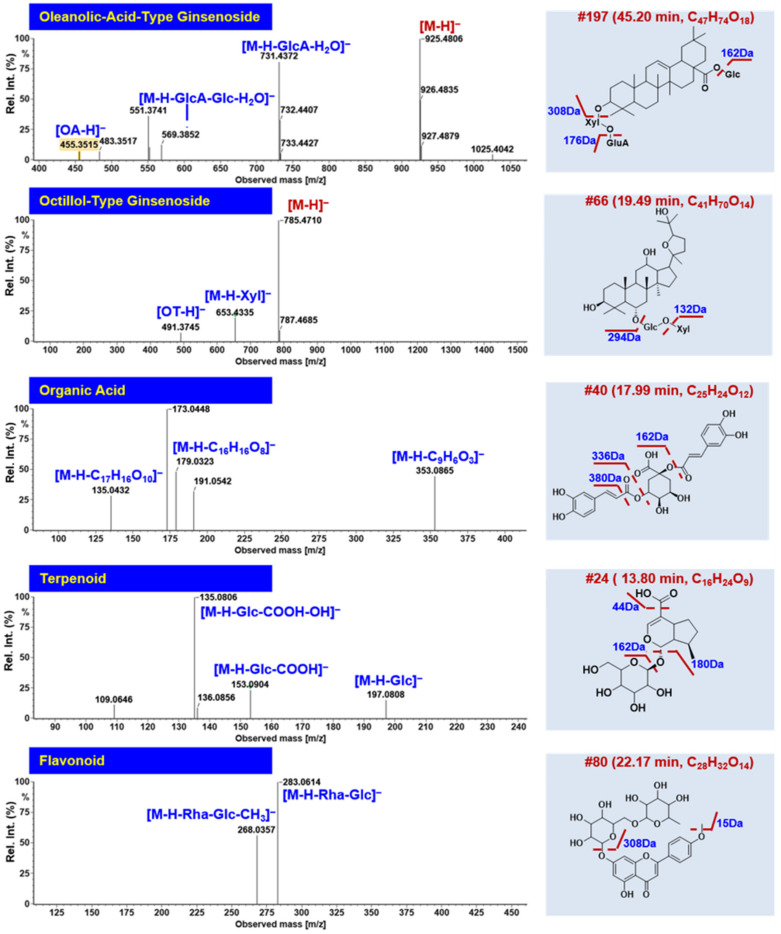
Annotation of the CID-MS^2^ spectra of the oleanolic-acid (OA) and octillol (OT)-type ginsenosides, organic acids, terpenoids, and flavonoids from WXG using UNIFI^TM^.

**Figure 8 molecules-27-03647-f008:**
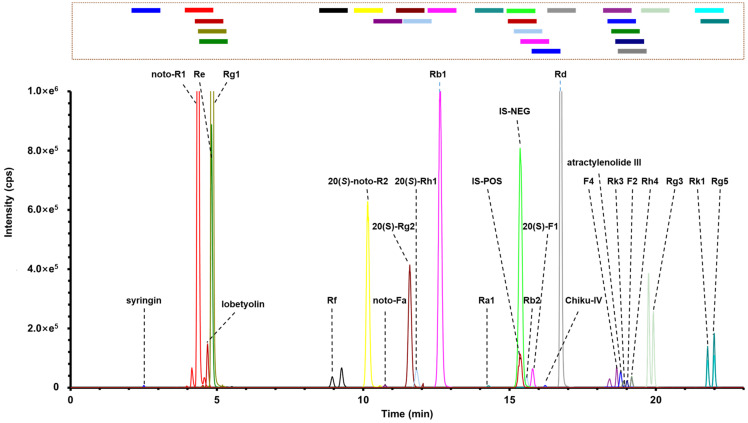
The sMRM chromatograms of 24 compounds obtained via the superposition of positive and negative ion modes.

**Figure 9 molecules-27-03647-f009:**
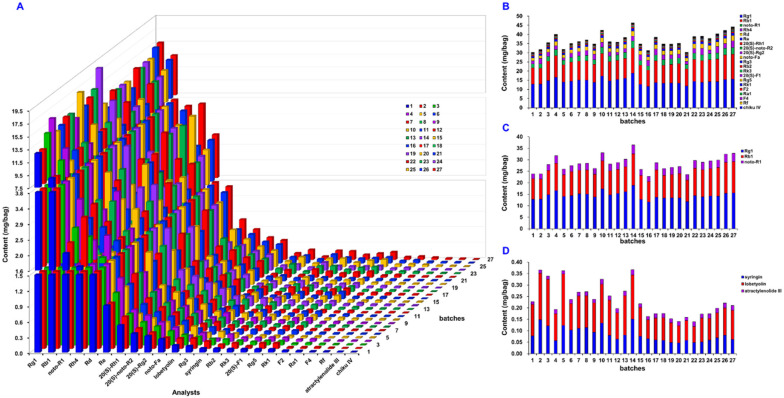
Content determination histograms of 24 compounds in 27 batches of commercial WXG samples: (**A**) the contents of all 24 compounds; (**B**) the contents of ginsenosides; (**C**) the contents of three ginsenoside markers recorded by the Chinese Pharmacopoeia; (**D**) the contents of three compounds originating from CP (*Codonopsis pilosula*).

**Table 1 molecules-27-03647-t001:** The sMRM parameters for the quantitative assay of 24 compounds from WXG.

No.	Compounds	Formula	Source	Key Parameters				
				Retention Time (min)	Q1 Mass (Da)	Q3 Mass (Da)	DP (V)	CE (V)
1	syringin	C_17_H_24_O_9_	CP	2.52	417.1	209.1	−48	−16
2	noto-R1	C_47_H_80_O_18_	PN	4.35	977.5	931.5	−94	−35
3	lobetyolin	C_20_H_28_O_8_	CP	4.67	441.2	215.1	−27	−16
4	Re	C_48_H_82_O_18_	PN	4.81	991.5	945.5	−101	−37
5	Rg1	C_42_H_72_O_14_	PN	4.82	845.5	799.5	−94	−32
6	Rf	C_42_H_72_O_14_	PN	8.9	845.5	799.5	−94	−32
7	20(*S*)-noto-R2	C_41_H_70_O_13_	PN	10.11	815.5	769.5	−109	−30
8	noto-Fa	C_59_H_100_O_27_	PN	10.67	1285.6	1239.6	−84	−39
9	20(*S*)-Rg2	C_42_H_72_O_13_	PN	11.54	829.5	783.5	−91	−33
10	20(*S*)-Rh1	C_36_H_62_O_9_	PN	11.77	683.4	637.4	−88	−29
11	Rb1	C_54_H_92_O_23_	PN	12.56	1153.6	1107.6	−99	−31
12	Ra1	C_58_H_98_O_26_	PN	14.22	1255.6	1209.6	−64	−35
13	Rb2	C_53_H_90_O_22_	PN	15.53	1123.6	1077.6	−108	−41
14	20(*S*)-F1	C_36_H_62_O_9_	PN	15.73	683.4	637.4	−88	−29
15	chiku-IV	C_47_H_74_O_18_	PN	16.17	925.5	569.4	−147	−60
16	Rd	C_48_H_82_O_18_	PN	16.71	991.5	945.5	−101	−37
17	F4	C_42_H_70_O_12_	PN	18.6	811.5	765.5	−69	−31
18	atractylenolide III	C_15_H_20_O_3_	CP	18.76	249.1	231.1	79	14
19	Rk3	C_36_H_60_O_8_	PN	18.85	665.4	619.4	−96	−27
20	F2	C_42_H_72_O_13_	PN	18.99	829.5	783.5	−91	−33
21	Rh4	C_36_H_60_O_8_	PN	19.13	665.4	619.4	−96	−27
22	Rg3	C_42_H_72_O_13_	PN	19.72	829.5	783.5	−91	−33
23	Rk1	C_42_H_70_O_12_	PN	21.73	811.5	765.5	−69	−31
24	Rg5	C_42_H_70_O_12_	PN	21.95	811.5	765.5	−69	−31
IS-Neg	astragaloside IV-NEG	C_41_H_68_O_14_	IS	15.31	829.5	783.4	−84	−39
IS-Pos	astragaloside IV-POS	C_41_H_68_O_14_	IS	15.33	785.5	473.4	123	17

Note: CP: *Codonopsis pilosula*; PN: *Panax notoginseng*; IS: internal standard.

## Data Availability

The data presented in this study are available in the article and Appendix A.

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
