# Peer review of "Systematic Qualitative and Quantitative Analyses of Wenxin Granule via Ultra-High Performance Liquid Chromatography Coupled with Ion Mobility Quadrupole Time-of-Flight Mass Spectrometry and Triple Quadrupole–Linear Ion Trap Mass Spectrometry"

_molecules, 2022, doi:10.3390/molecules27113647_

Round 1

Reviewer 1 Report

General comments

In the paper Systematic Qualitative and Quantitative Analyses of Wenxin Granule by Ultra-high Performance Liquid Chromatography Coupled with Ion Mobility Quadrupole Time-of-flight Mass Spectrometry and Triple Quadrupole/Linear Ion-trap Mass Spectrometry, authors report an interesting study on  multicomponent characterization and quantitative assay of WXG were conducted by developing two LC-MS approaches. 

Paper is interesting, hovewer several parts can be emproved. 

Specific comments

page 4 line 149 

25 to 40°C were accessed to observe the 

please add 25°

Method validation

The developed UHPLC-sMRM method was further validated in terms of selectivity, 386

linearity, sensitivity, precision, stability, repeatability, recovery, and matrix effect (Table 387

S3).

Please add a reference on method validation, for example you can us Online solid-phase extraction LC-MS/MS: A rapid and valid method for the determination of perfluorinated compounds at sub ng· L− 1 Level in natural water. Journal of Chemistry, 2018.

page 11 line 311

(R2) for 24 analytes was between 0.9913‒0.9990 over

Please note that R2 is determination coefficient

moreover, R2 = 0.9913 is very low.... at low or hight levels this can be produce e difference more than 30% in accuracy.

Page 16 

Regarding the ion source parameters, curtain gas and ion source gas 1 and gas 2 were maintained at 35, 55, and 55 psi, respectively; collision gas was medium; the source temperature 530 was set at 550°C;

For the nest study, please consider that several substances can be destroid at this temperature. 

I sugest, preliminary to investigated a T emperature of 300 °C

Author Response

Response to Reviewers’ Comments (molecules-1729015)

We are very grateful to the editor and three reviewers for providing the valuable and constructive comments/queries/suggestions to improve the quality of our manuscript. We generally agree with them, and thus carefully address all of these reviewers’ comments. The major revisions and new analyses we have undertaken are summarized below and discussed in detail in the point-by-point response. Hopefully, these revisions could well respond to these comments, and we look forward to the final acceptance and publication of this manuscript in Molecules.

Reviewers' comments

Reviewer #1:

General comments

In the paper, systematic Qualitative and Quantitative Analyses of Wenxin Granule by Ultra-high Performance Liquid Chromatography Coupled with Ion Mobility Quadrupole Time-of-flight Mass Spectrometry and Triple Quadrupole/Linear Ion-trap Mass Spectrometry, authors report an interesting study on multicomponent characterization and quantitative assay of WXG were conducted by developing two LC-MS approaches. Paper is interesting, however several parts can be improved.

Reply: We very much appreciate the reviewer for the efficient review work and the proposing of constructive suggestions to improve the quality of our manuscript. Our response to the comments is appended item by item as follows.

Specific comments

  • page 4 line 149, 25 to 40°C were accessed to observe the, please add 25°.

Reply: We are grateful to to this reviewer. According to the suggestion, we have revised “25 to 40°C” as “25°C to 40°C”, in the revised manuscript.

  • Method validation, the developed UHPLC-sMRM method was further validated in terms of selectivity, linearity, sensitivity, precision, stability, repeatability, recovery, and matrix effect (Table S3). Please add a reference on method validation, for example you can us Online solid-phase extraction LC-MS/MS: A rapid and valid method for the determination of perfluorinated compounds at sub ng· L− 1 Level in natural water. Journal of Chemistry, 2018.

Reply: We are grateful to the reviewer for the valuable comments and the reference provided. Following this suggestion, we have added the citation “Salvatore, B.; Maddalena, B.; Matteo, V.; Luisa, C.; Laura, C.; Pierluisa, D.; Amit, B. Online solid-phase extraction LC-MS/MS: a rapid and valid method for the determination of perfluorinated compounds at sub ng·L−1 level in natural water. J. Chem. 2018, 2018, 1-9.” as the reference [52].

  • Page 11 line 311, (R2) for 24 analytes was between 0.9913‒0.9990 over, Please note that R2 is determination coefficient moreover, R2 = 0.9913 is very low.... at low or high levels this can be produce e difference more than 30% in accuracy.

Reply: We thank the reviewer for the careful review work. Actually, the Analyst software gives the r (not R2) values (linear correlation coefficient). Therefore, in the revised manuscript, we changed R2 into r, with all the related expressions corrected in both the text and the Supporting Information. As a result, the r values for 24 analytes varied in the range of 0.9956-0.9995.  

  • Page 16, regarding the ion source parameters, curtain gas and ion source gas 1 and gas 2 were maintained at 35, 55, and 55 psi, respectively; collision gas was medium; the source temperature 530 was set at 550°C; For the nest study, please consider that several substances can be destroyed at this temperature. I suggest, preliminary to investigated a Temperature of 300 °C.

Reply: We thank the reviewer for this suggestion. When optimizing the ion pairs of 24 components, we used ion source temperature of 550℃, and all components gave good response either in the positive or negative mode. The service manual of QTrap 4500 recommend to use ion source temperature 550℃ at the flow rate of 0.3 mL/min. In other literature, we also found that the source temperature was set in the range of 500-600℃ to quantify ginsenosides (Quantification and discovery of quality control chemical markers for Ba-Bao-Dan by UPLC–MS/MS combined with chemometrics, Characterization and quantification of ginsenosides from the root of Panax quinquefolius L. by integrating untargeted metabolites and targeted analysis using UPLC-Triple TOF-MS coupled with UFLC-ESI-MS/MS). But we consider it’s a good suggestion to optimize this parameter to avoid the possible in-source decay for the analytes.

Again, we thank the reviewer for the efficient review work! 

Reviewer 2 Report

In my opinion, the paper is interesting and rigorous. Once is conveniently revised, I think it deserves to be published. Anyway, the manuscript is very long so it could overwhelm readers.

Many readers will not be familiar with this product, so I suggest adding introductory information with explanations of the potentially active components of these preparations and their importance/mechanism of action to treat the arrhythmias for which they are prescribed. The vast majority of the references are from Eastern authors, so I wondered if these types of products have a market in Eastern countries. Some comments on the commercial value of these products would be welcome

General points:

The criterion for the election of the analytical column is unclear. Figure 3 is quite useless to visualize the best separation. Thus, roughly, I think that other columns can be equally appropriate. Perhaps this figure could be moved to the supplementary section, and a new one could be drawn showing only the results of three or four of the best columns. Then, we will be able to understand their choice.

Besides, this section could be reinforced with some comments on the principal column features according to the nature of the stationary phase and their possible implications on the separation.

The elution gradient selected is complex and comprises several steps. Nothing is commented on how this gradient has been optimized. It is really necessary such a complex option.

The presentation of quantitative results as in figure 8a does not allow visualizing the information from minor species. For such purpose, I encourage the authors, not necessarily in this paper, to treat the information they have with chemometric methods to obtain a more satisfactory and comprehensive description of the behavior of the different samples.

Specific comments. 

Abstract. Define the second approach in the same way as it is the first one (e.g., UHPLC-QTRAP-MS).

Some figures should be split. For instance, Fig 3a and b are unconnected (the same for Fig. 7).

Some axis labels are missing, and others are unreadable.

The insets in Fig. 4 have not been explained in the figure caption.

The information depicted in Fig 7b could be visualized better with a bar plot displaying the extraction of some exemplary peaks/compounds.

Author Response

Response to Reviewers’ Comments (molecules-1729015)

We are very grateful to the editor and three reviewers for providing the valuable and constructive comments/queries/suggestions to improve the quality of our manuscript. We generally agree with them, and thus carefully address all of these reviewers’ comments. The major revisions and new analyses we have undertaken are summarized below and discussed in detail in the point-by-point response. Hopefully, these revisions could well respond to these comments, and we look forward to the final acceptance and publication of this manuscript in Molecules.

Reviewers' comments

Reviewer #2:

In my opinion, the paper is interesting and rigorous. Once is conveniently revised, I think it deserves to be published. Anyway, the manuscript is very long so it could overwhelm readers. Many readers will not be familiar with this product, so I suggest adding introductory information with explanations of the potentially active components of these preparations and their importance/mechanism of action to treat the arrhythmias for which they are prescribed. The vast majority of the references are from Eastern authors, so I wondered if these types of products have a market in Eastern countries. Some comments on the commercial value of these products would be welcome

Reply: We very much thank the reviewer for the positive evaluation on our current manuscript. According to this comment, during the revision process, we have added information about the pharmacological studies of Wenxin Granule (WXG) in the third paragraph of the introduction section.

WXG is one of the representative new drugs of Buchang (BC) Pharmaceutical with independent intellectual property rights. According to the data from Menet (https://www.menet.com.cn/), the three exclusive patented varieties of BC Pharmaceutical (including WXG) together accounted for 8.83% of the market share of cardiovascular Chinese patent medicines in 2018, and the market share rankings all entered the top 20. The company's core products have strong market competitiveness. What’ more, In February 2019, the column “Modern Chinese Medicine” published the “Summary of Competitiveness Report on Science and Technology of Big Brand Traditional Chinese Medicine (TCM) (2019 Edition)”. The report shows that WXG is one of the top 10 products in the list of the top 100 products in the "Technology Factors of Big Brand TCM (All Categories)". At the same time, in the ranking list of scientific and technological factors of Big Brand TCM for cardiovascular diseases in the report, WXG ranks among the top three oral drugs for cardiovascular diseases. The above information can reflect the high importance of WXG in China in treating cardiovascular diseases.

General points:

  • The criterion for the election of the analytical column is unclear. Figure 3 is quite useless to visualize the best separation. Thus, roughly, I think that other columns can be equally appropriate. Perhaps this figure could be moved to the supplementary section, and a new one could be drawn showing only the results of three or four of the best columns. Then, we will be able to understand their choice.

Reply: We thank the reviewer very much for this comment. The original Figure 3 has been changed in the revised manuscript. The comparison results of representative four stationary phase were shown in Figure 3 of the new version as appended below, and the detailed comparison results for 20 chromatographic columns were shown in Figure S1.

Besides, this section could be reinforced with some comments on the principal column features according to the nature of the stationary phase and their possible implications on the separation.

Reply: We thank the reviewer for proposing such good suggestions. The universal, silica-based bonded phase used for the HSS T3 sorbents is compatible with 100% aqueous mobile phase and can enhance retention of polar molecules, which is believed to have better separation for the polar components of WXG, such as ginsenosides and steroid saponins. That’s also a reason to select this column. We have added this content in the revised manuscript.

  • The elution gradient selected is complex and comprises several steps. Nothing is commented on how this gradient has been optimized. It is really necessary such a complex option.

Reply: We thank the reviewer for this comment. A better elution gradient is important for the compounds analysis. In this work, we adjusted the elution gradient using a three-stage method (the entire effective separation time period was divided into three sections): the front, the middle, and the back. It was optimized in segments from the front to the back. And the end point of elution gradient optimization is the chromatogram baseline is stable, the distribution of chromatographic peaks is balanced, and the resolution of key chromatographic peaks is satisfactory. We have added the related discussion in the text of the revised version.

  • The presentation of quantitative results as in figure 8a does not allow visualizing the information from minor species. For such purpose, I encourage the authors, not necessarily in this paper, to treat the information they have with chemometric methods to obtain a more satisfactory and comprehensive description of the behavior of the different samples.

Reply: We thank the reviewer for this valuable suggestion. As showed in Figure 8a, the content of different compounds among the multiple batches varied significantly. Therefore, we provided bar charts of Figure 8 B, C and D, to further visualize the quantitative results of ginsenosides, three Chinese Pharmacopoeia ginsenoside markers, and three compounds of CP. Moreover, the contents of minor components in 27 batches of WXG samples can be found in Table S6.

We agree with this good suggestion, and in our future work, we shall utilize the chemometrics to visualize the overall batch-to-batch quality difference of the TCM we analyze.

Specific comments.

  • Define the second approach in the same way as it is the first one (e.g., UHPLC-QTRAP-MS).

Reply: We thank the reviewer for this valuable comment. In abstract, the quantitative method has been modified as “ultra-high performance liquid chromatography/triple quadrupole-linear ion trap mass spectrometry (UHPLC/QTrap-MS)”.

  • Some figures should be split. For instance, Fig 3a and b are unconnected (the same for Fig. 7).

Reply: We thank the reviewer for this comment. The original Figure 3 has been divided into two parts: Figures 3 and 4, in the revised manuscript. The same was Figure 7, in which the pooled sMRM chromatograms were retained, but the extraction condition optimization was moved to the Supporting Information as Figure S7.

  • Some axis labels are missing, and others are unreadable.

Reply: We thank the reviewer for the careful review work. For Figure 7, we have divided it into two parts. The lower part regarding the extraction optimization has been changed into Figure S7 by the way of bar charts, in which the axis and axis labels have been added (Supporting Information).

  • The insets in Fig. 4 have not been explained in the figure caption.

Reply: We deeply appreciate the reviewer’s suggestion. Following this suggestion. we have enriched the figure caption of the original Figure 4 (Newly numbered as Figure 5): Figure 5. Base peak intensity (BPI) chromatogram of WXG in both the positive and negative ESI modes. Those peaks confirmatively characterized with the aid of reference compounds are annotated in red. Peaks 98‒104, 106, 108‒112, 115‒118, 127‒131, 148, 171, 178‒182, 185‒187, and 189, are marked separately in the upper right corner for the better observation, in the revised manuscript.

  • The information depicted in Fig 7b could be visualized better with a bar plot displaying the extraction of some exemplary peaks/compounds.

Reply: We appreciate the reviewer for the careful review work. To solve this issue, we have changed the original chromatogram to the bar plot, which has been moved to the Supporting Information as Figure S7. We optimized the extraction solvent by comparing the peak area values of 24 analytes, and the results showed methanol had good extraction effect on the most of the ginsenosides.

Again, we thank the reviewer for the efficient review work!

Reviewer 3 Report

This is a very detailed UHPLC-MS study of wenxin granule, a traditional Chinese medicine. As indicated by the authors and confirmed by a Google Scholar search, there are very few previous such studies. Although the MS study is very thorough, was optimization of the sample preparation emphasized enough? A comparison of acetonitrile and methanol as extraction solvents was made but were time and temperature considered?This could of course affect the type of compounds isolated. A comparison of the compounds identified to the main previous study cited in reference 25 should be provided in the Discussion section. Some of the figures such as Figure 3 are hard to read even when expanded by x150 on the computer. Figure 3 and perhaps others need to be simplified and some of the plots placed in Supplementary Information.

Author Response

Response to Reviewers’ Comments (molecules-1729015)

We are very grateful to the editor and three reviewers for providing the valuable and constructive comments/queries/suggestions to improve the quality of our manuscript. We generally agree with them, and thus carefully address all of these reviewers’ comments. The major revisions and new analyses we have undertaken are summarized below and discussed in detail in the point-by-point response. Hopefully, these revisions could well respond to these comments, and we look forward to the final acceptance and publication of this manuscript in Molecules.

Reviewers' comments

Reviewer #3:

This is a very detailed UHPLC-MS study of wenxin granule, a traditional Chinese medicine. As indicated by the authors and confirmed by a Google Scholar search, there are very few previous such studies. Although the MS study is very thorough, was optimization of the sample preparation emphasized enough? A comparison of acetonitrile and methanol as extraction solvents was made but were time and temperature considered? This could of course affect the type of compounds isolated.

Reply: We thank the reviewer for the careful review work. We agree on the point that, the extraction temperature and extraction time may affect the type of compound that can be extracted. In this work, the ultrasonic extraction we selected was repeated twice, and the total extraction time was 2 h under the optimized condition. Compared with the third time of extraction (total extraction 3 h), all analytes for the quantitative analysis could be of high extraction rate by the ultrasonic extraction for two times. Moreover, considering some components may be unstable under the exposure at high temperature, such as malonyl-ginsenoside Rb2 and malonyl-ginsenoside Re, we thus performed the mild ultrasonic extraction (400 W, 40 kHz) at low temperature of 25°C.

A comparison of the compounds identified to the main previous study cited in reference 25 should be provided in the Discussion section.

Reply: We thank the reviewer for this comment. Compared with the reference 25 (newly numbered as reference 29), our study identified 205 chemical components from WXG in total, while the reference 29 identified 68 chemical components. Besides, our study newly reported the identification of alkaloids. We have added the discussion by comparison with the reference, in the revised version of the manuscript.

Some of the figures such as Figure 3 are hard to read even when expanded by x150 on the computer. Figure 3 and perhaps others need to be simplified and some of the plots placed in Supplementary Information.

Reply: We thank the reviewer for this comment. The content regarding the 20 columns comparison has been divided into two parts in the revised manuscript. The comparison results of four representative stationary phases are shown in Figure 3, while the detailed comparison results of 20 chromatographic columns is shown in Figure S1, as the Supporting Information.

Again, we are grateful to the reviewer for the efforts in commenting on our manuscript!